# Percutaneous vertebroplasty versus percutaneous kyphoplasty for osteoporotic vertebral compression fractures: an umbrella review protocol of systematic reviews and meta-analyses

Qingyang Gao,[1] Qiujiang Li ![ORCID],[2] Liang Wang,[2] Ying Cen,[1] Huiliang Yang ![ORCID][2]

QG, QL and LW contributed equally.

¹Department of Plastic and Burn Surgery, West China Hospital, Sichuan University, Chengdu, Sichuan, China
²Department of Orthopedics, Orthopedic Research Institute, West China Hospital, Sichuan University, Chengdu, Sichuan, China

**Correspondence to**
Dr Huiliang Yang;
huiliang_yang@wchscu.cn

## ABSTRACT

**Introduction** Several systematic reviews and meta-analyses have confirmed that percutaneous vertebroplasty and percutaneous kyphoplasty showed safety and beneficial efficacy in patients with osteoporotic vertebral compression fractures. Whereas, there is wide variation among results, which are not conducive to the evaluation and use of clinicians. This study will investigate the efficacy and safety of percutaneous vertebroplasty and percutaneous kyphoplasty for the treatment of osteoporotic vertebral compression fractures, aiming to provide a more reliable evidence base for clinical practice in treating osteoporotic vertebral compression fractures.

**Methods and analysis** We will retrieve the relevant articles using the five databases(PubMed, Scopus, EMBASE, Cochrane Library and Web of Science) from inception to March 2023 for systematic review and meta-analysis comparing the overall safety and efficacy of percutaneous vertebroplasty and percutaneous kyphoplasty in patients with osteoporotic vertebral compression fractures. Three reviewers will screen citation titles, abstracts and evaluate the full text of each relevant citation based on prespecified eligibility criteria. Any discrepancies in decisions between reviewers will be resolved through discussion. We will assess the methodological quality of the included studies according to A MeaSurement Tool to Assess systematic Reviews 2 checklist.

**Ethics and dissemination** This umbrella review will inform clinical and policy decisions regarding the benefits and harms of percutaneous vertebroplasty versus percutaneous kyphoplasty for osteoporotic vertebral compression fractures. Neither primary data nor individual patient information will be collected, thus ethics approval is not required. Findings will be reported through a peer-reviewed publication, conference presentations and the popular press.

**PROSPERO registration number** CRD42021268141.

## STRENGTHS AND LIMITATIONS OF THIS STUDY

⇒ This research will conduct a comprehensive review of both meta-analyses and systematic reviews to assess the reliability of current evidence on percutaneous vertebroplasty in the management of osteoporotic vertebral compression fractures compared with percutaneous kyphoplasty.

⇒ Screening, data collection and quality assessment were independently conducted by two individuals during the study.

⇒ Each included review will undergo quality assessment and evidence grading as part of the study protocol.

⇒ Only literature published in English will be included.

## INTRODUCTION

As the society ages, osteoporotic vertebral compression fractures (OVCFs) caused by osteoporosis have become a major worldwide health issue.[1 2] OVCFs often occur as a result of low-impact injuries and patients may not have a clear history of trauma. They primarily present with persistent back pain and local vertebral kyphosis, which significantly affects the survival rates and living standards of the elderly population.[3–6]

Non-surgical therapies for OVCFs include bed rest and drug therapy.[7 8] However, these methods often provide inadequate pain relief, leaving patients with limitations in their daily activities. Moreover, complications such as pressure sores, pneumonia, deep vein thrombosis and urinary tract infections may arise during the treatment. Surgical interventions are typically reserved for more severe fractures.[9] Currently, percutaneous vertebroplasty (PVP) and percutaneous kyphoplasty (PKP) have been widely applied in OVCFs treatment. Both techniques offer immediate pain relief and help stabilise the fractured vertebral body.[10–14] Specifically, PVP involves restoring the vertebral volume by injecting polymethylmethacrylate bone cement through a minimally invasive wound directly into the affected area.[15] PKP represents an advancement over PVP. By placing a balloon

into the involved vertebral body before the injection, PKP creates a regularly shaped cavity to control the distribution of the bone cement.[15] Consequently, some researchers believe that PKP should carry a lower risk of bone cement leakage and provide more stable reinforcement than PVP.[16–18]

Nonetheless, the debate persists regarding which method, PKP or PVP, is superior for treating OVCFs. In a meta-analysis of six randomised clinical trial (RCT) studies involving 1077 patients, Zhu et al[19] performed a meta-analysis of 6 RCT studies involving a total of 1077 patients and finally found that both PKP and PVP could improve function and relieve pain, but PKP had lower cement leaking rates. By contrast, a meta-analysis of 16 studies aiming to evaluate the efficacy and safety of PKP/PVP for the treatment of OVCFs found that PKP could better reduce bone cement leakage, restore vertebral height restoration and correct vertebral angulation in comparison to PVP.[20] The same conclusion is drawn by Chang et al.[21]

Systematic reviews and meta-analyses (SRs/MAs) are important research methods in evidence-based medicine and the best sources of clinical evidence[22] to create clinical guidelines and develop recommendations for medical practice. Although there are multiple published SRs/MAs studies on PKP/PVP for OVCFs, they produced varying and sometimes conflicting results, which could impact clinical decision-making.

To provide a comprehensive overview of these SRs/MAs studies, an umbrella review (or systematic review of systematic reviews) is needed. As another important part of evidence-based medicine, an umbrella review assesses the methodological and report quality of included SRs/MAs and provides the quality of evidence. This will be the first study to summarise SRs/MAs on clinical outcomes of PVP and PKP for the treatment of OVCFs. An umbrella review presented in our protocol will identify, collect, evaluate and integrate the existing knowledge of PKP/PVP for OVCFs to determine the more beneficial surgical treatments between PKP and PVP for patients with OVCFs and guide optimal treatment strategies. The results of this study will provide evidence for clinicians and further guide researchers to embark on relevant studies.

## METHODS AND ANALYSIS
### Study registration
The study protocol was registered on the International Prospective Register of Systematic Reviews (https://www.crd.york.ac.uk/prospero) with a unique ID of CRD42021268141. It was designed using the methodology guidelines for umbrella reviews provided by the Joanna Briggs Institute (JBI).[23] Meanwhile, this protocol will follow the guidance of the Preferred Reporting Items for Systematic Review and Meta-analysis Protocols[24] (see online supplemental appendix 1 for the checklist).

The research questions of this umbrella review were as follows:

1. What is the efficacy (primary outcomes) of PKP (intervention) compared with PVP (comparison) in OVCFs patients (population)?
2. What is the safety (secondary outcomes) of PKP (intervention) compared with PVP (comparison) in OVCFs patients (population)?
3. To identify the preoperative and peri-operative factors (intervention and comparison) that affect the functional improvements and radiographic improvements (outcome) in OVCFs patients (population).

### Patient and public involvement
No patients or the public will be involved in this study.

### Search strategy
An electronic search will be conducted among various databases, including PubMed, Scopus, EMBASE, Cochrane Library and Web of Science with the Medical Subject Headings (MeSH) and keywords "percutaneous vertebroplasty", "percutaneous kyphoplasty", "balloon kyphoplasty", "osteoporosis vertebral compression fractures", "systematic reviews", "meta-analysis". The terms will be put into combinations with Boolean search syntax operators to retrieve the maximum number of articles.

As a supplement, the 'related article' function from PubMed will be used to further identify potential articles that were eligible for inclusion in the review. In addition, grey literature and Google Scholar will also be searched. To include as many relevant studies as possible, we will consult with content experts. A draft search strategy for PubMed is provided in table 1.

### Eligibility criteria
#### Type of studies
We will include both SRs/MAs studies comparing different clinical outcomes following PKP/PVP.

The studies should report the outcome data separately for the PKP and PVP groups.

#### Publication time
The date range will be set from database inception to March 2023.

#### Language of literature
The language of publication is restricted to English.

### Inclusion criteria
We set the inclusion criteria for this umbrella review according to the population, intervention, comparison, outcome principle.[25]

#### Population
Patients with VCFs due to osteoporosis will be included. We impose a lower age limit ≥60 years to identify the elderly. Expect that, there was no restriction on gender, race or healthy status.

#### Intervention
Interventions include PVP with curative intent

| Table 1 | Search strategy for PubMed/MEDLINE |
|---|---|
| #1 | "kyphoplasty"[mh] OR "balloon vertebroplasty"[all] OR "vertebroplasty, balloon"[all] |
| #2 | "vertebroplasty"[mh] |
| #3 | #1 AND #2 |
| #4 | (((((((percutaneous vertebroplast*[Title/Abstract]) OR pereutancous vertebroplast*[Title/Abstract]) OR vertebroplast*[Title/Abstract]) OR cementoplast*[Title/Abstract]) OR sacroplast*[Title/Abstract]) OR balloon kyphoplast*[Title/Abstract]) OR kyphoplast*[Title/Abstract]) |
| #5 | #3 OR #4 |
| #6 | "spinal fractures"[mh] OR "fracture, spinal"[all] OR "fractures, spinal"[all] OR "spinal fracture"[all] |
| #7 | "fractures, compression"[mh] OR "compression fracture"[all] OR "fracture, compression"[all] OR "compression fractures"[all] |
| #8 | #6 AND #7 |
| #9 | (spinal compression fracture[Title/Abstract]) OR vertebral compression fracture[Title/Abstract] |
| #10 | #8 OR #9 |
| #11 | "meta-analysis"[pt] OR "systematic review"[pt] OR "meta-analysis as topic"[mh] OR "systematic reviews as topic"[mh] OR cochrane*[all] OR evidence-based review*[all] OR health technology assessment*[all] OR meta analy*[all] OR meta-review*[all] OR meta-synthes*[all] OR research overview*[all] OR systematic overview*[all] OR systematic review*[all] OR metaanaly*[all] OR meta-analy*[all] OR metanaly*[all] |
| #12 | #5 AND #10 AND #11 |

## Comparison

PKP with curative intent is considered as comparison.

The volume of injected cement, fluoroscopy times, intraoperative blood loss and operation time were recorded for both intervention and comparison groups.

## Outcomes
### Primary outcomes include
1. Functional outcomes: the Visual Analogue Scale score, Oswestry Disability Index, Short Form-36.
2. Radiographic outcomes included Cobb angle and anterior vertebral body height.

### Secondary outcomes include
1. Bone cement leakage.
2. New VCFs after the treatment.

All outcomes will be assessed based on the definitions applied in the selected meta-analysis. Studies will not be included or excluded on the basis of reported outcomes.

## Data collection
### Study selection
First, literature not meeting the eligibility criteria will be excluded through the screening of titles and abstracts. Subsequently, the full text of preliminarily selected articles will be independently reviewed by two reviewers (QG and QL) to confirm the inclusion and exclusion criteria for enrolled items. Using a data extraction form, two additional reviewers (WL and YC) will extract data, with subsequent cross-checking by the initial reviewers (QG and QL). If there is a disagreement in the checked results, the arbitration will be resolved through discussion with a third reviewer (HY). The flow chart of the literature selection process is shown in figure 1.

### Data extraction
Two reviewers (WL and YC) will extract data independently based on a tool developed by the JBI Data Extraction form. When data extraction fails to reach an agreement, the arbitration will be resolved through discussion with a third reviewer (HY). If data were depicted in graphs or incomplete, we will try to contact authors to obtain original data by email.

Extracted data include:
1. Basic information of the literature: first author, publication year range, country.
2. Information of the study: study design included, sample size, search strategy, quality evaluation method, original research involved and interest disclosure.
3. Population: gender, age, race, pain and deformity before treatment.
4. Intervention/comparison: treatment, level of the vertebra, the volume of injected cement, fluoroscopy times, intraoperative blood loss and operation time.
5. Outcomes: primary outcome and secondary outcome as described above, time point to obtain the outcomes.
6. Conclusion.

For meta-analysis, we will extract summary effect sizes (random effect size and/or fixed effect size, 95% CIs) and significance levels. When accessible, information on between-study heterogeneity (Cochrane Q statistic or $I^2$) and small-study effects will also be gathered.

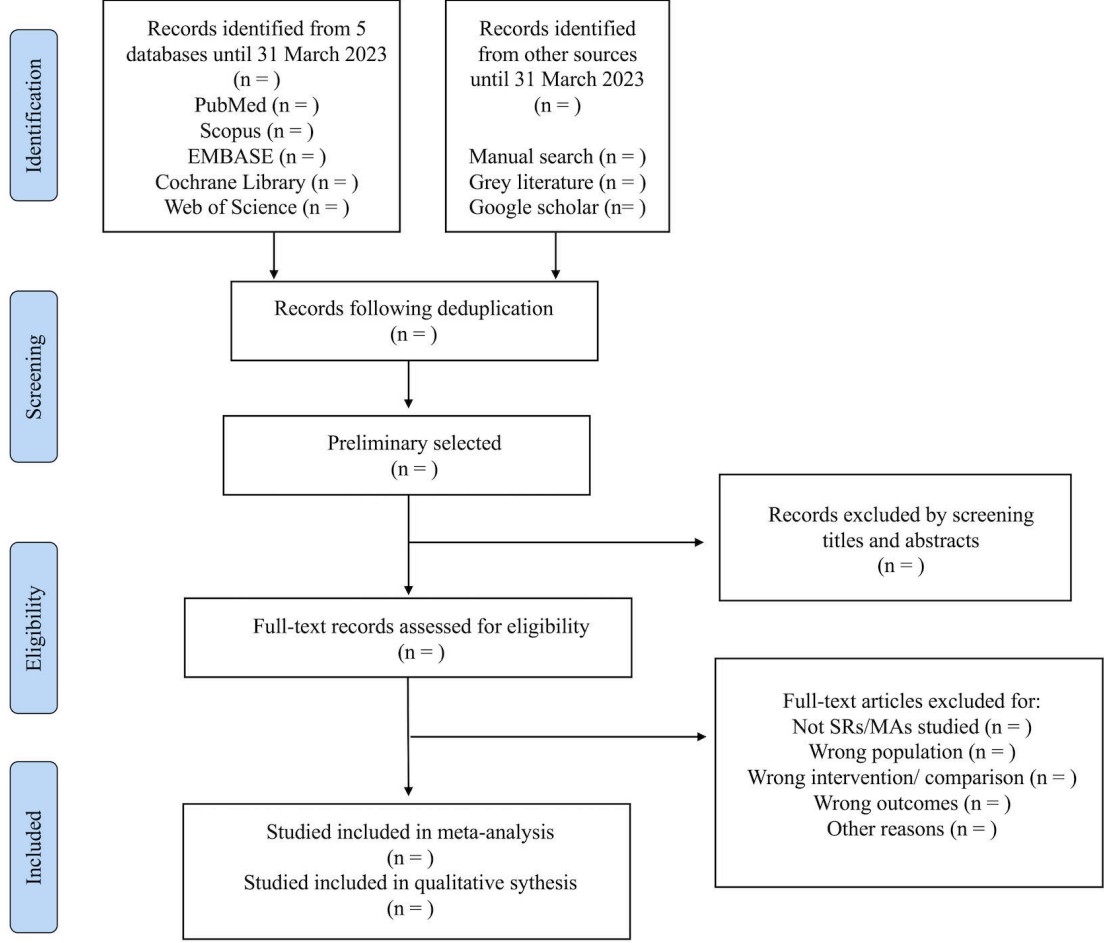

**Figure 1** PRISMA flow chart of study selection process. PRISMA, Preferred Reporting Items for Systematic Reviews and Meta-Analyses; SRs/MAs, systematic reviews and meta-analyses.

## Data synthesis and statistical analysis

First, tables and graphs will be used to delineate the characteristics of all eligible studies and provide a concise summary of their findings, guided by the Cochrane Handbook.[26] Then, for meta-analysis, further statistical analysis will be conducted. Using both fixed and random effects models, the pooled effect size, 95% CI, p value and of the selected meta-analyses will be calculated. Significance levels will be categorised into three thresholds ($p \leq 0.05$, $p \leq 10^{-3}$ and $p \leq 10^{-6}$). Given the potential overlap of individual studies within each meta-analysis, the outcomes will be interpreted with consideration for duplication. Heterogeneity among the included studies will be presented as Cochran's Q statistics and the $I^2$ index. Egger regression asymmetry test will be used to show the presence of the small study effects. The methodological quality of the meta-analysis will be evaluated using the A Measurement Tool to Assess Systematic Reviews 2 (AMSTAR 2) tool.[27]

The relationship and findings between studies will be explored through a narrative synthesis. In the concluding phase of the integrative review, we will articulate and summarise the key findings and conclusions of the studies. Subgroup analysis, taking into account the influence of variables such as sex, age and outcomes, will be

also conducted. The statistical analysis will be performed by using STATA (V.17).

### Identifying the overlap of the literature

Given the inevitable duplication of original research in our selected review, we will employ the method outlined by Wegner *et al*[28] to visualise the duplication. In brief, a table will be created with the names of systematic reviews displayed as column headers, and the individual studies from each review listed as row entries. Then, a formula was used to compute the corrected covered area (CCA)[29]:

$$CCA = \frac{N-r}{r(c-r)}$$

Here, N represents the total number of included original publications in the reviews, r denotes the number of single studies of each review (rows) and c signifies the number of systematic reviews (columns).

For those CCA scores >11%, our approach will involve including the most recently published and highest quality review, while excluding other overlapping literature that is homogeneous. Conversely, literature with a CCA score <5% will prompt a meticulous examination of the search strategy and research scope. CCA scores falling between

5% and 11% will be regarded as indicative of a moderate level of overlap.

## Identifying possible publication bias

Reporting bias will be assessed through a funnel plot. To mitigate the potential impact of subjective visual observation, Begg's and Egger's tests will be conducted to evaluate the asymmetry of the funnel plot, as a supplementary approach.

## Methodological quality evaluation

Two reviewers independently assess the methodological quality of systematic reviews based on AMSTAR 2.[27 29 30] When quality assessment results fail to reach an agreement, arbitration will be sought through discussion with a third reviewer. The AMSTAR 2 is a reliable and valid critical appraisal tool designed for systematic reviews, applied to both randomised and non-randomised studies of healthcare interventions (see online supplemental appendix 2 for the checklist) . AMSTAR 2 comprises 16 items, with 7 identified as critical domains (items 2, 4, 7, 9, 11, 13, 15).[30] Further details can be found in online supplemental appendix 3. The results of evidence quality for systematic reviews will be categorised into four levels according to AMSTAR 2: critically low, low, medium and high.

## Grading the evidence

According to the previous studies,[31–33] the evidence for outcomes in each meta-analysis was classified as follows:

Convincing (class I): P value of the random-effect model $<10^{-6}$, case number >1000, low heterogeneity between the studies, non-significant bias or small study effect, robustness to unmeasured confounding, significant result (p<0.05) for the largest study, 95% prediction interval not including the null value and $I^2<50\%$.

Highly suggestive (class II): P value of the random-effect model $<10^{-6}$, case number >1000 (or more than 20 000 participants for continuous outcomes) and significant result (p<0.05) for the largest study.

Suggestive (class III): P value of the random-effect model $<10^{-3}$ and case number >1000.

Weak (class IV): Evidence was assigned to the remaining significant association with a p value of the random-effect model <0.05;

Non-significant (class V): P value for the random effect model >0.05.

**Contributors** All authors made substantive intellectual contributions in this study to qualify as authors. QG and QL developed the research question. WL and QL registered the protocol and wrote the first draft of the manuscript. QG and HY developed the search strategy and created the data extraction form. YC and HY provided critical review and feedback at each stage of the process. All authors were involved in writing the manuscript. All authors read and approved the final manuscript.

**Funding** This study was funded by The Science and Technology Project of the Health Planning Committee of Sichuan (No. 21PJ036) and the National Natural Science Foundation of China (No. 82102521).

**Competing interests** None declared.

**Patient and public involvement** Patients and/or the public were not involved in the design, or conduct, or reporting, or dissemination plans of this research.

**Patient consent for publication** Not applicable.

**Provenance and peer review** Not commissioned; externally peer reviewed.

**ORCID iDs**
Qiujiang Li http://orcid.org/0009-0001-3331-3674
Huiliang Yang http://orcid.org/0000-0002-0182-4234

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
