## [Reviewer comments · BMJ Open]

ARTICLE DETAILS

TITLE (PROVISIONAL)	Percutaneous Vertebroplasty vs Percutaneous Kyphoplasty for Osteoporotic Vertebral Compression Fractures: An Umbrella Review Protocol of Systematic Reviews and Meta-Analyses
AUTHORS	Gao, Qingyang; Li, Qiujiang; Liang, Wang; Cen, Ying; Yang, Huiliang

VERSION 1 – REVIEW

REVIEWER	Jiayong Liu The University of Toledo Medical Center
REVIEW RETURNED	25-May-2023

GENERAL COMMENTS	The protocol of this umbrella review was designed to investigate the efficacy and safety of PVP vs PKP for patients with OVCFs. It followed Joanna Briggs Institute's guidelines. The study design and methodology were standard. It could prevent unnecessary duplication of work and will hopefully enable potential collaboration.
---

REVIEWER	Hongwei Wang General Hospital of Shenyang Military Area Command of Chinese PLA, Department of Orthopedics
REVIEW RETURNED	12-Aug-2023

GENERAL COMMENTS	Reject. I don't know why you chose the title "Percutaneous Vertebroplasty vs Percutaneous Kyphoplasty for Osteoporotic Vertebral Compression Fractures". The study was not innovative, why not limit it to one research point such as unilateral puncture, navigation system.
---

REVIEWER	Amir Mohammad Salehi Hamadan University of Medical Sciences
REVIEW RETURNED	03-Nov-2023

GENERAL COMMENTS	The method of data analysis is unclear. 1. Authors should specify with which software they performed the analysis. 2. Expand your statistical tests further. 3. One of the most important parts of umbrella studies is grading the evidence, I have not seen any explanation about this analysis. To strengthen the study methodology, the authors can use these studies: https://link.springer.com/article/10.1186/s13643-022-01915-6 https://www.sciencedirect.com/science/article/abs/pii/S2468784722001994
---

REVIEWER	Dimitrios Tsartsalis Linköping University Hospital, Department of Clinical Physiology
REVIEW RETURNED	05-Nov-2023

GENERAL COMMENTS	This is a study protocol aiming to investigate the efficacy and safety of percutaneous vertebroplasty and kyphoplasty in treating osteoporotic vertebral compression fractures, and evaluate the existing evidence of these procedures by performing an umbrella review of systematic reviews and meta-analyses. The manuscript addresses an interesting topic in clinical terms, and the proposed methodology is sound and appropriate to answer the research questions. Nonetheless, there are some issues that should be taken into account prior to consideration for publication:  1. The introduction is well-written and straightforward. However, as the manuscript deals with specific, advanced procedures in a specialised clinical area, I think that a brief description of these techniques should facilitate the paper's visibility and broaden its clinical interest. 2. In the Methods section-eligibility criteria-type of studies, the authors mention that they will include systematic reviews and meta-analyses. I think that a more thorough description of potentially eligible studies should be provided. For instance, are the authors going to include both meta-analyses of prospective and retrospective studies? meta-analyses of RCTs? 3. In the same section, in publication time, the search of databases for eligible articles is limited up to 31st March 2023. As this is only the initial study protocol (before data extraction, analysis etc), I think that the time limit should not be so strict. 4. Which method are the authors going to use to identify duplicate meta-analyses? 5. The main purpose for conducting an umbrella review is to evaluate the evidence of published data in a specific area. Which method will the authors use to grade the strength of evidence of published meta-analyses in their study? 6. Some minor grammatical comments: try to be coherent regarding the time of the verbs used in the text. As it is a protocol for a planned research project, I think that past tenses should not be used when describing data extraction, synthesis and statistical analysis.
------------------	--

VERSION 1 – AUTHOR RESPONSE

Reviewer 1

1. The protocol of this umbrella review was designed to investigate the efficacy and safety of PVP vs PKP for patients with OVCFs. It followed Joanna Briggs Institute's guidelines. The study design and methodology were standard. It could prevent unnecessary duplication of work and will hopefully enable potential collaboration.

Thank you for this positive comment. We hope that our research will contribute valuable insights to the ongoing debate on the optimal treatment strategies for OVCF patients.

Reviewer 2

2.Reject. I don't know why you chose the title "Percutaneous Vertebroplasty vs Percutaneous Kyphoplasty for Osteoporotic Vertebral Compression Fractures". The study was not

innovative, why not limit it to one research point such as unilateral puncture, navigation system.

Thank you for your advice. Indeed, there are lots of original research and meta-analyses aimed at unraveling the efficacy and safety of PKP and PVP. However, there's still a controversy over the superior method for treating OVCF patients. On one hand, the quality of the literature impacts the credibility of their conclusions. On the other hand, our goal is to identify confounding factors and sources of heterogeneity, which could assist in further subdividing the applicable population or in improving the surgical procedure. We are optimistic that the findings of this study will furnish surgeons with evidence to make informed decisions when selecting optimal treatment strategies.

Reviewer 3

1. Authors should specify with which software they performed the analysis.

Thank you for your recommendation. We have now incorporated details about the statistical software used into the manuscript.

(Page 7/ Line 13) "The statistical analysis will be performed by using STATA (version 17)."

2. Expand your statistical tests further.

Thank you for your advice, we have made modifications in accordance with this suggestion. And here is the revision:

(Page 6/ Line 31-Page 7/Line 13)"Firstly, tables and graphs will be used to delineate the characteristics of all eligible studies and provide a concise summary of their findings, guided by the Cochrane Handbook [26]. Then, for meta-analysis, further statistical analysis will be conducted. Utilizing both fixed and random effects models, the pooled effect size, 95% CI, p-value, and of the selected meta-analyses will be calculated. Significance levels will be categorized into three thresholds ($p \leq 0.05$, $p \leq 10^{-3}$, and $p \leq 10^{-6}$). Given the potential overlap of individual studies within each meta-analysis, the outcomes will be interpreted with consideration for duplication. Heterogeneity among the included studies will be presented as Cochran's Q statistics and the I^2 index. Egger regression asymmetry test will be used to show the presence of the small study effects. The methodological quality of the meta-analysis will be evaluated using the A Measurement Tool to Assess Systematic Reviews 2 (AMSTAR 2) tool [27]. The relationship and findings between studies will be explored through a narrative synthesis. In the concluding phase of the integrative review, we will articulate and summarize the key findings and conclusions of the studies. Subgroup analysis, taking into account the influence of variables such as sex, age, and outcomes, will be also conducted. The statistical analysis will be performed by using STATA (version 17)."

3. One of the most important parts of umbrella studies is grading the evidence, I have not seen any explanation about this analysis.

Thanks for your careful checks. We are sorry for our carelessness. The description of evidence grading has been added to the method section.

(Page 8/ Line 9-22)"According to the previous studies [31–33], the evidence for outcomes in each meta-analysis was classified as follows: Convincing (class I): p-value of the random-effect model < 10^{-6} , case number > 1000, low heterogeneity between the studies, non-significant bias or small study effect, robustness to unmeasured confounding, significant result ($p < 0.05$) for the largest study, 95% prediction interval not including the null value, and $I^2 < 50\%$; Highly suggestive (class II): p-value of the random-effect model < 10^{-6} , case number > 1000 (or more than 20,000 participants for

continuous outcomes), and significant result ($p < 0.05$) for the largest study; Suggestive (class III): p-value of the random-effect model $< 10^{-3}$, and case number > 1000 ; Weak (class IV): evidence was assigned to the remaining significant association with a p-value of the random-effect model < 0.05 ; Non-Significant (Class V): p-value for the random effect model > 0.05 .”

Reviewer 4

This is a study protocol aiming to investigate the efficacy and safety of percutaneous vertebroplasty and kyphoplasty in treating osteoporotic vertebral compression fractures, and evaluate the existing evidence of these procedures by performing an umbrella review of systematic reviews and meta-analyses. The manuscript address an interesting topic in clinical terms, and the proposed methodology is sound and appropriate to answer the research questions.

Nonetheless, there are some issues that should be taken in account prior to consideration for publication:

1. The introduction is well-written and straightforward. However, as the manuscript deals with specific, advanced procedures in a specialised clinical area, I think that a brief description of these techniques should facilitate the paper’s visibility and broaden its clinical interest.

Thank you for your advice, we have added a brief introduction to the specialized techniques in this section.

(Page 3/ Line 1-10) “Currently, percutaneous vertebroplasty (PVP) and percutaneous kyphoplasty (PKP) have been widely applied in OVCFs treatment. Both techniques offer immediate pain relief and help stabilize the fractured vertebral body [10–14]. Specifically, PVP involves restoring the vertebral volume by injecting polymethylmethacrylate (PMMA) bone cement through a minimally invasive wound directly into the affected area[15]. PKP represents an advancement over PVP. By placing a balloon into the involved vertebral body before the injection, PKP creates a regularly shaped cavity to control the distribution of the bone cement[15]. Consequently, some researchers believe that PKP should carry a lower risk of bone cement leakage and provide more stable reinforcement than PVP[16–18].”

2. In the Methods section-eligibility criteria-type of studies, the authors mention that they will include systematic reviews and meta-analyses. I think that a more thorough description of potentially eligible studies should be provided. For instance, are the authors going to include both meta-analyses of prospective and retrospective studies? meta-analyses of RCTs?

Thank you for your advice. In our revision, we have made sure to include both systematic reviews and meta-analysis studies in our research. This emphasis is intended to enhance the comprehensiveness of our study and provide a more thorough exploration of the available evidence.

(Page 4/ Line 30-31) “We will include both systematic reviews and meta-analysis studies comparing different clinical outcomes following PKP/PVP..”

3. In the same section, in publication time, the search of databases for eligible articles is limited up to 31st March 2023. As this is only the initial study protocol (before data extraction, analysis etc), I think that the time limit should not be so strict.

This section is revised according to reviewer comment.

(Page 5/ Line 5)“The dates range will be set from database inception to March, 2023.”

4. Which method are the authors going to use to identify duplicate meta-analyses?

We plan to use corrected covered area (CCA) statistics to identify the duplication of meta-analyses. Subsequently, we will judiciously eliminate redundant research to ensure the integrity and uniqueness of our study. This approach aims to enhance the precision and reliability of our findings. If you have any additional insights or recommendations, we welcome your input.

(Page 7/ Line 15-28) “Given the inevitable duplication of original research in our selected review, we will employ the method outlined by Wegner et. Al[28] to visualize the duplication. In brief, a table will be created with the names of systematic reviews displayed as column headers, and the individual studies from each review listed as row entries. Then, a formula was used to compute the corrected covered area (CCA)[29]:

$$CCA = \frac{N - r}{r(c - r)}$$

Here, N represents the total number of included original publications in the reviews, r denotes the number of single studies of each review (rows), and c signifies the number of systematic reviews (columns).

For those CCA scores >11%, our approach will involve including the most recently published and highest quality review, while excluding other overlapping literature that is homogeneous. Conversely, literature with a CCA score < 5% will prompt a meticulous examination of the search strategy and research scope. CCA scores falling between 5% to 11% will be regarded as indicative of a moderate level of overlap.”

5. The main purpose for conducting an umbrella review is to evaluate the evidence of published data in a specific area. Which method will the authors use to grade the strength of evidence of published meta-analyses in their study?

We sincerely apologize for our oversight. In response to this, we have included a detailed description of evidence grading in the methodology section. The revision is as follows:

(Page 8/ Line 9-22)“According to the previous studies [31–33], the evidence for outcomes in each meta-analysis was classified as follows: Convincing (class I): p-value of the random-effect model < 10⁻⁶, case number > 1000, low heterogeneity between the studies, non-significant bias or small study effect, robustness to unmeasured confounding, significant result (p < 0.05) for the largest study, 95% prediction interval not including the null value, and I² < 50%; Highly suggestive (class II): p-value of the random-effect model < 10⁻⁶, case number > 1000 (or more than 20,000 participants for continuous outcomes), and significant result (p < 0.05) for the largest study; Suggestive (class III): p-value of the random-effect model < 10⁻³, and case number > 1000; Weak (class IV): evidence was assigned to the remaining significant association with a p-value of the random-effect model < 0.05; Non-Significant (Class V): p-value for the random effect model > 0.05.”

6. Some minor grammatical comments: try to be coherent regarding the time of the verbs used in the text. As it is a protocol for a planned research project, I think that past tenses should not be used when describing data extraction, synthesis and statistical analysis.

Thank you for your acknowledgment. We have systematically revised the method section to ensure uniformity in verb tense, opting for the future tense throughout.

VE 14-Dec-2023RSION 2 – REVIEW

REVIEWER	Dimitrios Tsartsalis Linköping University Hospital, Department of Clinical Physiology
REVIEW RETURNED	14-Dec-2023
GENERAL COMMENTS	Accept. The authors responded properly in all comments.